# Mixed Prototype Correction for Causal Inference in Medical Image Classification

**Yajie Zhang**
Department of Computing
The Hong Kong Polytechnic
University
Hong Kong SAR, China
yajie.zhang@connect.polyu.hk

**Zhi-An Huang***
Research Office
City University of Hong Kong
(Dongguan)
Dongguan, China
huang.za@cityu-dg.edu.cn

**Zhiliang Hong**
Shengli Clinical Medical College
Fujian Medical University
Fujian, China
Ultrasound Department
Fuzhou University Affiliated
Provincial Hospital
Fuzhou, China
601615603@qq.com

**Songsong Wu***
Shengli Clinical Medical College
Fujian Medical University
Fujian, China
Ultrasound Department
Fuzhou University Affiliated
Provincial Hospital
Fuzhou, China
465269898@qq.com

**Jibin Wu**
Department of Data Science and
Artificial Intelligence, and
Department of Computing
The Hong Kong Polytechnic
University
Hong Kong SAR, China
jibin.wu@polyu.edu.hk

**Kay Chen Tan***
Department of Data Science and
Artificial Intelligence
The Hong Kong Polytechnic
University
Hong Kong SAR, China
kctan@polyu.edu.hk

## Abstract

The heterogeneity of medical images poses significant challenges to accurate disease diagnosis. To tackle this issue, the impact of such heterogeneity on the causal relationship between image features and diagnostic labels should be incorporated into model design, which however remains underexplored. In this paper, we propose a mixed prototype correction for causal inference (MPCCI) method, aimed at mitigating the impact of unseen confounding factors on the causal relationships between medical images and disease labels, so as to enhance the diagnostic accuracy of deep learning models. The MPCCI comprises a causal inference component based on front-door adjustment and an adaptive training strategy. The causal inference component employs a multi-view feature extraction (MVFE) module to establish mediators, and a mixed prototype correction (MPC) module to execute causal interventions. Moreover, the adaptive training strategy incorporates both information purity and maturity metrics to maintain stable model training. Experimental evaluations on four medical image datasets, encompassing CT and ultrasound modalities, demonstrate the superior diagnostic accuracy and reliability of the proposed MPCCI. The code will be available at https://github.com/Yajie-Zhang/MPCCI.

*Corresponding author.

## CCS Concepts

• **Computing methodologies** → *Image representations.*

## Keywords

Disease diagnosis, Causal inference, Front-door adjustment, Multi-view prototype learning

**ACM Reference Format:**
Yajie Zhang, Zhi-An Huang, Zhiliang Hong, Songsong Wu, Jibin Wu, and Kay Chen Tan. 2024. Mixed Prototype Correction for Causal Inference in Medical Image Classification. In *Proceedings of the 32nd ACM International Conference on Multimedia (MM '24), October 28-November 1, 2024, Melbourne, VIC, Australia.* ACM, New York, NY, USA, 10 pages. https://doi.org/10.1145/3664647.3681395

## 1 Introduction

Medical image classification provides essential support to the clinicians and other medical professionals in diagnosing and treating patients by analyzing lesion features of the human body within medical images [11]. With wide applications in the real world, this problem has been extensively researched. In the past few decades, especially driven by the application of deep learning technologies, a significant body of methods have been developed, which can generally be categorized into detection-based approaches [10, 32], segmentation-based approaches [28, 44, 45] and feature extraction approaches [22, 23, 29, 30]. Despite the remarkable advancements in previous studies, classifying images in the medical domain remains much more challenging than it is for natural images. This is primarily attributed to the inherently complex nature of the medical images. Compared with the natural images, the medical images often contain more noise and artifacts, due to the limitations of current imaging technologies, such as weak X-ray penetration of the

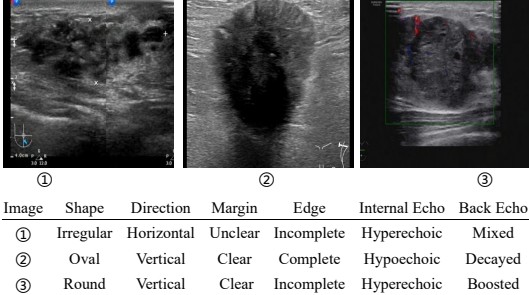

| Image | Shape | Direction | Margin | Edge | Internal Echo | Back Echo |
|---|---|---|---|---|---|---|
| ① | Irregular | Horizontal | Unclear | Incomplete | Hyperechoic | Mixed |
| ② | Oval | Vertical | Clear | Complete | Hypoechoic | Decayed |
| ③ | Round | Vertical | Clear | Incomplete | Hyperechoic | Boosted |

**Figure 1: Illustration of different manifestations for the same type of lesion in breast cancer ultrasound images. Lesion attributes are also provided in the table below images.**

equipment, presence of gas in the body, and motion artifacts [12]. In addition to these two factors that have received more attention in previous research, lesion heterogeneity is also a crucial challenge to the classifiers, which however is much less exploited.

Lesion heterogeneity in the medical imaging refers to the variability in features and appearances of the same disease, encompassing variations in terms of shape, size, density, intensity, texture, and other lesion characteristics. An exemplar illustration is given in Fig. 1. As shown in the figure, breast cancer may exhibit pronounced heterogeneity in ultrasound images, with lesions varying significantly in appearance. For example, the lesion shapes may be oval, round, or irregular, and the complexity is further exacerbated when combined with other varied attributes as shown in the table below the images. The impact of heterogeneity on model performance has been acknowledged in previous studies [27, 49], and there have been some attempts to alleviate its negative effects through variance pooling structures [5] and data augmentation [52]. However, these methods often yield suboptimal outcomes due to the lack of a thorough examination on the root causes of heterogeneity.

Lesion heterogeneity is caused by various factors, such as the diverse origins of cancer cells, variable gene expressions, and patient-specific susceptibilities [3]. These factors have significant influences upon the prediction of a diagnostic label from the medical image. For example, the patients with genetic predispositions are more susceptible to the illness [3]. Hence in this work, we propose to model these factors by leveraging causal inference [37] for enhanced medical image classification. To enunciate our idea, we establish a structural causal model for the medical image classification task, as shown in Fig. 2. In this figure, the underlying causes of heterogeneity, denoted as C, involve various factors as above mentioned; the medical images X and the diagnostic outcomes Y are both impacted by the heterogeneity cause factors C. In formulation, there exist X → Y, denoting the causal path that image X contains the lesion representations related to the given label Y, and X ← C → Y representing the backdoor path that X and Y exhibit spurious correlation through C. That is, the factors in C affect the characterization of medical imaging (X ← C); in addition, these factors influence the probability of a patient contracting a particular disease (C → Y). Following the Pearl's causal inference theory [37], when we try to find the causal effect of X on Y, we want the nodes we condition on to block any "backdoor" path in which one end has an arrow to X,

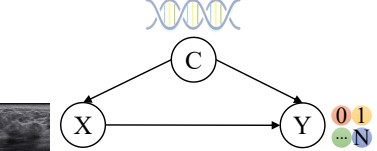

**Figure 2: The structural causal model for a disease with heterogeneity. C represents the cause of heterogeneity, X denotes medical images, and Y represents diagnostic results.**

because such paths may make X and Y dependent, but are obviously not transmitting causal influences from X; and if we do not block them, they will confound the effect that X has on Y. Therefore, we should adjust the confounders "C" to block the backdoor path for better inference from X to Y. However, given the inherent difficulty or even impossibility of quantifying these confounders, their adverse impact upon the image-based diagnostic procedures remains elusive and unaddressed.

In this work, we propose a novel approach for enhanced medical image classification, named mixed prototype correction for causal inference (MPCCI), which mitigates the influences of the confounding factors on the medical diagnosis by exploiting front-door adjustment (FDA) [37]. The FDA introduces a mediator variable, denoted as A in Fig. 3, between the causal path of variables X and Y, to adjust the causal pathway between them, which well addresses the immeasurability of the elusive confounding factors. To implement MPCCI, we first design a multi-view feature extraction (MVFE) module with spatial-channel attention that allows these multi-view features to serve as mediators in FDA to link the causal effect from images to labels. We also develop a mixed prototype correction (MPC) module that exchanges some features of the multi-view features and the multi-view prototypes to effectively apply causal intervention on the mediators. The multi-view prototypes contain meta-knowledge of various disease categories, and the causal intervention mechanism that exchanges features with them can correct the spurious association between X and Y formed by the confounders. To improve the smoothing of the feature exchange process, an adaptive training strategy is presented, comprising two key components: information purity (IP) and maturity (MT). The IP module is used to measure the proportion of noise in the feature exchange process, and MT is used to measure the stability of the model to noise in different training stages.

Experiments on four medical datasets verify the effectiveness of the proposed MPCCI on diagnosing covid, breast cancer, lymph node metastasis, and thyroid. In summary, the contributions of this work are as followings:

- This work is the first attempt to conduct cause-effect analysis to alleviate the immeasurable condounders for enhancing medical image classification. The proposed method solves the problem by applying an FDA strategy, treating the multi-view features as mediators to infer the causalities from images to labels.
- The proposed MPCCI includes two key modules (MVFE and MPC) to achieve FDA step by step, which effectively mitigates the adverse effects of the confounders upon medical

diagnosis. An adaptive training strategy, consisting of IP and MT modules, is introduced to mitigate the noise effect in the MPC module.
- The proposed MPCCI exhibits promising performance across four distinct disease diagnosis tasks, yielding dependable interpretability results.

## 2 Related Work

### 2.1 Medical Image Classification

Currently, medical image classification, a task aiming to identify disease categories from unseen medical images, is generally tackled by training deep learning models over annotated training datasets. The model performance is mainly dependent on its architecture design as well as the scale of the training data. Some methods adopt advanced architectures, such as AlexNet [26], ResNet [20], VGG [47], and ViT [32] for good classification performance. In recent year, some works propose to extract and integrate multi-scale information to improve classification accuracy, utilizing feature pyramid networks [45], dilated convolutions [14], and attention mechanisms [19, 30], etc. There are also models fusing local and global features [8, 31] to achieve lifted efficacy in medical image classification. These methods are all based on the assumptions of sufficient high-quality training data, which are not always true. In addition to these model-centric approaches, some other methods broaden the diversity and volume of training data to enhance model generalization by utilizing GANs [4], variational autoencoders [6], MixUp [13], and diffusion models [25]. These data-centric approaches also demonstrate strong effectiveness on medical image classification tasks. However, either the model-centric models, or the data-centric ones, fall short on addressing disease heterogeneity and its related confounding factors, which hampers their performance.

### 2.2 Causal Inference

The goal of causal inference is to unravel the complex causal relationships between variables, far beyond the mere correlations [50]. It serves as a powerful tool for understanding the roots and implications of phenomena and thereby supports informed decision-making and interventions. For its potent analytical power, causal inference has been applied in various domains, such as medical image classification [38], domain generalization [36], and medical image segmentation [7, 34]. In medical image analysis, some methods [7, 34, 36] treat complex organ co-occurrences and background phenomena such as pseudo artifacts as observable confounding factors, and leverage the backdoor adjustment strategy [37] for causal intervention. Some works [38] harness counterfactual reasoning for medical image analysis by crafting counterfactual samples to neutralize the effects of observable confounding factors. These causal inference based methods achieve promising results. Yet, they tend to focus on observable confounding factors, which constrains their effectiveness in handling cases with unobservable confounders. In this work, we propose to utilize the FDA strategy to mitigate the impact of unmeasured confounders for better medical image classification performance.

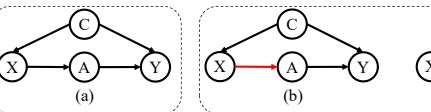

**Figure 3: (a) A structural causal model for medical image classification. (b) (c) Two steps of FDA, representing calculations of $P(A|do(X))$ and $P(Y|do(A))$ (red lines). Red fork denotes the causal intervention from X to A.**

## 3 Cause-Effect Analysis

In this section, we provide a brief analysis of the causal relationships among the elements in our tasks, namely the input image $X$, multi-view features $A$, image label $Y$, and confounders $C$, using a structural causal model (SCM) illustrated in Fig. 3 (a). We also describe how FDA is used in this context.

The main causal relationships in Fig. 3 (a) include $X \to A \to Y$, $C \to X$, and $C \to Y$. 1) For $X \to A \to Y$, the input image $X$ is fed into a deep neural network to extract multi-view features $A$, which are then used to predict the label $Y$. 2) For $C \to X$, the confounders $C$ like genetics, origins of cancer cells, patient habits, etc. influence the lesion manifestation $X$. 3) For $C \to Y$, the confounders $C$ can also affect the disease category $Y$ of a patient. For example, the patients with genetic predisposition for breast cancer have a higher risk of malignancy.

Note that there are two paths connecting $X$ and $Y$: the front-door path $X \to A \to Y$ and the backdoor path $X \leftarrow C \to Y$. The existence of the backdoor path makes it difficult to evaluate the true causality from $X$ to $Y$ through deep networks. If $C$ is measurable, the backdoor adjustment can be used to eliminate the link of $C \leftarrow X$. However, since most of $C$ in this work are not measurable, we turn to use front-door adjustment [37] to estimate the causality from $X$ to $Y$. To achieve this, FDA employs a mediator $A$ to transmit knowledge of $X$ to $Y$ through the front-door path, and then evaluates the causalities from $X$ to $Y$ by combining the causal effects of $X$ to $A$ and $A$ to $Y$, i.e., to estimate the probabilities $P(A|do(X))$ and $P(Y|do(A))$, respectively. The $do$-operation represents an active intervention to a cause rather than a passive observation.

The $\mathbf{P(A|do(X))}$ represents the causal relationship between $X$ and $A$, as illustrated in Fig. 3 (b). Since the path of $X \leftarrow C \to Y \leftarrow A$ is blocked by the collider [37], we can write

$$P(A|do(X)) = P(A = a|X = x). \tag{1}$$

The $\mathbf{P(Y|do(A))}$ (Fig. 3 (c)) pursues the true causality between $A$ and $Y$ without confounders $C$. There are two paths from $A$ to $Y$: $A \to Y$ and the backdoor path $A \leftarrow X \leftarrow C \to Y$. Due to the existence of the backdoor path, we need to cut off the link between $A$ and $X$ by controlling $X$, and we can write $P(Y|do(A))$ as

$$P(Y|do(A)) = \sum_x P(Y = y|A = a, X = x). \tag{2}$$

Through layer-by-layer causal effect calculation, the causality from $X$ to $Y$ can be represented as

$$P(Y|do(X)) = \sum_a P(Y = y|do(A = a))P(A = a|do(X = x))$$
$$= \sum_a P(A = a|X = x) \sum_{x'} P(Y = y|A = a, X = x')P(X = x'), \tag{3}$$

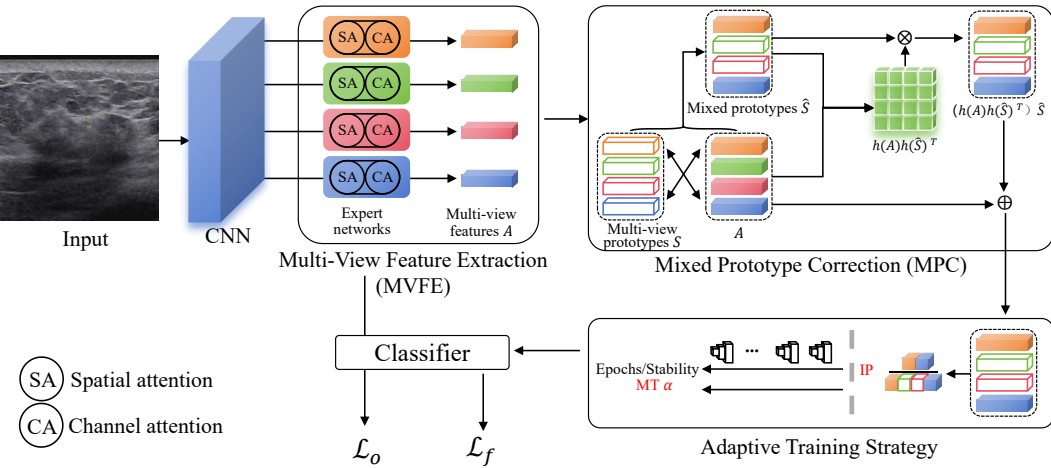

**Figure 4: Illustration of the MPCCI framework. MPCCI consists of three main components: the MVFE, MPC, and the adaptive training strategy. MVFE involves expert networks that use spatial-channel attention to generate multi-view features. MPC is implemented by fusing mixed prototypes with original multi-view features to simulate $\sum_{x'} P(Y|A, x')$. In addition, the adaptive training strategy, consisting of IP and MT, is adopted to improve the smoothing of the feature exchange process.**

where $x'$ is an index of summation in $P(Y|do(A))$.

## 4 Methodology

In this section, we introduce the proposed Mixed Prototype Correction for Causal Inference (MPCCI) approach in medical image classification. As shown in Fig. 4, it involves multi-view feature extraction (MVFE) and mixed prototype correction (MPC) modules to implement the FDA strategy. Additionally, we present an adaptive training approach, incorporating the information purity (IP) and the maturity (MT), to alleviate the noise at MPC. The IP module quantifies the noise proportion during the feature exchange process, while MT assesses the model's robustness to noise across various training phases.

### 4.1 Multi-View Feature Extraction

The MVFE module is responsible for generating multi-view features $A$, which serve as the mediator in Fig. 3 (a) and are used to implement $P(A|do(X))$ in Eq. (1). First, we input the image into a convolutional neural network (CNN) such as ResNet18 [18] to obtain feature maps $\mathbf{E} = f_b(x) \in \mathbb{R}^{D \times H \times W}$, where $f_b$ is the function of the CNN, and $D$, $H$, $W$ represent the number of channels, height, and width of $\mathbf{E}$, respectively. To extract multi-view features from the $\mathbf{E}$, we employ two parallel paths. We apply global average pooling to $\mathbf{E}$ to obtain a global feature vector $\mathbf{g} \in \mathbb{R}^D$ and construct expert networks [21] with spatial-channel attention following CBAM [43]. Spatial-channel attention adopted in CNN allows for the adaptive weighting of feature maps across both spatial and channel dimensions. This enables the network to selectively focus on informative features, enhancing its ability to learn and identify complex visual patterns. To ensure that each expert network learns different features of an image, they are initialized with different parameters. The formulation of the expert networks can be expressed as

$$\mathbf{a}^k = f_a^k(\mathbf{E}) \in \mathbb{R}^D, \tag{4}$$

where $f_a^k$ represents the $k$-view feature vector $\mathbf{a}^k$ generated by the $k$-th function of spatial-channel attention.

After extracting the mediator $\mathbf{A}$, it is crucial to ensure that the learned multi-view features are distinct across classes. To achieve this goal, the multi-view features and global feature are concatenated and then fed into the classifier (a fully-connected layer is used in this work) denoted as $f_c$, to produce the predicted label $\mathbf{y}$ of the image $x$:

$$\mathbf{y} = f_c(\mathbf{g}, \mathbf{A}) = f_c(\mathbf{g}||\mathbf{a}^1|| \cdots ||\mathbf{a}^K) \in \mathbb{R}^C, \tag{5}$$

where $||$ represents the concatenation operation, $C$ is the number of categories, and $K$ is the number of expert networks. The cross-entropy loss is used to optimize $f_c$:

$$\mathcal{L}_o = - \sum l_c log \frac{exp(y_c)}{\sum_{j=1}^{C} exp(y_j)}, \tag{6}$$

where $l \in \mathbb{R}^C$ denotes the ground-truth label of $x$. If $x$ belongs to the $c$-th category, $l_c = 1$; otherwise $l_c = 0$.

### 4.2 Mixed Prototype Correction

The MPC module aims to correct the side-effects of confounders and further explore the causality of $A$ on $Y$ by estimating

$$P(Y|do(A)) = \sum_{x'} P(Y|A, x')P(x'). \tag{7}$$

However, it is infeasible to collect all possible $x'$ (lesions that might appear in reality) with $A$ for predicting $Y$. Thus we use mixed multi-view prototypes to approximate $x'$. Multi-view prototype learning [48] is an emerging machine learning technique that aims to learn a set of prototypes across different views to capture the underlying structure of representative examples for each category. Specifically, we use the $c$-th class-specific average multi-view features to approximate the $c$-th multi-view prototypes, denoted as $\mathbf{S}^c = \{\mathbf{s}_1^c, \cdots, \mathbf{s}_K^c\} \in \mathbb{R}^{K \times D}$. We then generate the mixed multi-view

prototypes $\hat{S}$, which partially come from the source multi-view prototypes ($c$-th) and another random counterparts ($c'$-th), to express $x'$ as

$$\hat{S} = \mathbf{v}S^c + (\mathbf{1} - \mathbf{v})S^{c'}, \tag{8}$$

where $\mathbf{v} \in \{0, 1\}^K$ represents the random exchanging index vector of $S^c$. Namely, each $v_k$ represents whether the $k$-th prototype in $S^c$ should be exchanged by the $k$-th prototype in $S^{c'}$. Since $\hat{S}$ captures lesions that have distinct characteristics in specific view features, it can serve as a substitute for $x'$. To predict label $Y$, we fuse $\hat{S}$ with $\mathbf{A}$ via a fusion module. Cross-attention [30] is a mechanism that enables neural networks to capture the interdependent relationship between two heterogeneous features using a learnable similarity matrix. In this work, we incorporate cross-attention with the feature mapping function $h(\cdot)$ into the fusion module to explore this independence. The output is fused multi-view features denoted as $\hat{\mathbf{A}}$:

$$\hat{\mathbf{A}} = \mathbf{A} + (h(\mathbf{A})h(\hat{S})^T)\hat{S}. \tag{9}$$

We can concatenate $\hat{\mathbf{A}}$ with global feature vector $\mathbf{g}$ to predict label as $\hat{y} = f_c(\mathbf{g}, \hat{\mathbf{A}})$ using Eq. (5).

## 4.3 Adaptive Training Strategy

We utilize an adaptive training strategy to maintain stable model training. Since $\hat{S}$ randomly mixes the $c$-th and $c'$-th multi-view prototypes, there is a possibility that $\hat{\mathbf{A}}$ is predicted as the $c'$-th category. In this study, we hypothesize that two factors are related to this situation: 1) the information purity (IP) in $\hat{S}$, and 2) the maturity (MT) of the fusion module.

The IP refers to the amount of prototype information from the source category contained in $\hat{S}$. If $\hat{S}$ contains a large amount of prototype information from other categories, the probability of $\hat{\mathbf{A}}$ being predicted as other categories increases. Therefore, we use $\frac{\sum \mathbf{v}}{K}$ and $\frac{\sum (1-\mathbf{v})}{K}$ to represent the possibility of $\hat{\mathbf{A}}$ being predicted as the $c$-th class and $c'$-th class, respectively.

The MT represents the ability of the fusion module to accurately fuse label-related prototype information to $\hat{\mathbf{A}}$. When the fusion module cannot fuse multi-view prototypes well, the probability of accurately predicting $\hat{y}_c$ decreases. We assume that MT increases with the process of network iterative optimization, and we denote MT as $\alpha = \alpha_0 + \frac{cur\_epoch}{total\_epoch}$, where $\alpha_0$ represents the initialized maturity. Based on these two factors, we can write the probabilities of $\hat{y}$ to be $\hat{y}_c$ and $\hat{y}_{c'}$:

$$P(\hat{y}_c) = min(1, \frac{\sum \mathbf{v}}{K} + \alpha); P(\hat{y}_{c'}) = max(0, \frac{\sum (1 - \mathbf{v})}{K} - \alpha). \tag{10}$$

Based on this adaptive training strategy, the optimization goal of MPC can be formulated as:

$$\begin{aligned}
\mathcal{L}_f = - \sum_{x'} (P(\hat{y}_c) l_c log \frac{exp(\hat{y}_c)}{\sum_{j=1}^C exp(\hat{y}_j)} \\
+ P(\hat{y}_{c'}) l_{c'} log \frac{exp(\hat{y}_{c'})}{\sum_{j=1}^C exp(\hat{y}_j)}) P(x'),
\end{aligned} \tag{11}$$

where $P(x')$ is set to a uniform distribution $\frac{1}{N}$ because $x'$ is generated from a random mixture with equal probability.

## 4.4 Overall Loss Function

By combining the MVFE, MPC, and the adaptive training strategy, the overall loss function $\mathcal{L}$, which is the optimization objective to be minimized during training iterations, is a combination of the original loss $\mathcal{L}_o$ and the fusion loss $\mathcal{L}_f$:

$$\mathcal{L} = \mathcal{L}_o + \lambda \mathcal{L}_f, \tag{12}$$

where $\lambda$ is a hyperparameter that controls the relative weight of the fusion loss. To ease the understanding of MPCCI, the pseudo-code is presented as **Algorithm 1**.

---

**Algorithm 1** The pseudo-code of MPCCI

---

**Input:** Training dataset $X = \{(x_1, l_1), (X_2, l_2), \cdots, (x_n, l_n)\}$
**Outpt:** Predicted labels
 1: Initialize network of MPCCI:
      $f = f_c(f_b(\cdot)||f_a^1(f_b(\cdot))|| \cdots ||f_a^K(f_b(\cdot)))$
 2: **for** i = 1, ..., Epoch **do**
 3:   /* MVFE */
 4:   $\mathbf{E} \leftarrow f_b(x)$ // Compute feature maps
 5:   $\mathbf{a}^k \leftarrow f_a^k(\mathbf{E})$ // Compute mutli-view features
 6:   $\mathbf{g} \leftarrow avgpool(\mathbf{E})$ // Compute the global feature
 7:   $\mathbf{y} \leftarrow f_c(\mathbf{g}, \mathbf{A})$ // Compute the predicted label
 8:   Compute $\mathcal{L}_o$ via Eq. (6)
 9:   /* MPC */
10:   $S_k^c \leftarrow \frac{1}{n_c}(\mathbf{a}_{k1}^c + \cdots + \mathbf{a}_{kn_c}^c)$ // The multi-view prototypes
11:   $\hat{S} \leftarrow \mathbf{v}S^c + (\mathbf{1} - \mathbf{v})S^{c'}$ // The mixed prototypes
12:   $\hat{\mathbf{A}} \leftarrow \mathbf{A} + (h(\mathbf{A})h(\hat{S})^T)\hat{S}$ // The fused multi-view features
13:   $\hat{y} \leftarrow f_c(\mathbf{g}, \hat{\mathbf{A}})$
14:   /* The adaptive training strategy */
15:   Compute the IP factor: $\frac{\sum \mathbf{v}}{K}$ and $\frac{\sum (1-\mathbf{v})}{K}$
      // The probabilities to be predicted as $c$-th and $c'$-th classes
16:   $\alpha \leftarrow \alpha_0 + \frac{cur\_epoch}{total\_epoch}$ // The MT
17:   Compute $\mathcal{L}_f$ via Eq. (11)
18:   Update $f$ by minimizing $\mathcal{L} = \mathcal{L}_o + \lambda \mathcal{L}_f$
19: **end for**

---

## 5 Experiment

We conduct comprehensive experiments to evaluate the performance of the porposed MPCCI approach. At below, we first introduce the datasets used for experiments, evaluation protocols, compared methods, and implementation details. Then, we report and analyze the quantitative results obtained across four medical datasets. Moreover, the validation of heterogeneity cause C and data set analysis are conducted to systematically evaluate MPCCI. We also take further experimental analysis to assess the capabilities of MPCCI. This analysis encompasses an examination of the number of features, the function of the mixing mechanism, and the visualization results.

## 5.1 Datasets

Four medical image datasets are utilized for the evaluation of MPCCI. The details of the datasets are provided as follows:

The **CT COVID-19** [33] dataset comprises 7,593 COVID-19 CT images sourced from 466 patients, 6,893 normal CT images

from 604 patients, and 2,618 CAP CT images from 60 patients. For experimentation purposes, a total of 14,486 images from the normal and COVID-19 categories are selected. These images are randomly partitioned into training, validation, and test sets at a ratio of 7:1:2.

The **BUSI** [2] is a publicly available dataset consisting of 780 ultrasound images of three classes, i.e., normal, benign, and malignant. In this work, we follow the setting of MIB Net [42] which achieves the best result reported to date and use only the benign images (437) and malignant images (210). Conforming to the protocol of MIB Net, the dataset is partitioned into training, validation, and test sets at a ratio of 8:1:1.

The **FJPH** is a dataset established by ourselves for predicting the likelihood of lymph node metastasis. The data inside are obtained anonymously from a local hospital (Fujian Provincial Hospital) to ensure the privacy of all involved patients. It consists of 889 ultrasound images categorized into two classes: metastasis (500 ultrasound images) and non-metastasis (389 ultrasound images). We employ this dataset to demonstrate the versatility of the proposed method in different ultrasound imaging scenarios. Adhering to the settings of BUSI, we partition this dataset into training, validation, and test sets at a ratio of 8:1:1.

The **FJTU** is a thyroid ultrasound dataset established from the Fujian Provincial Hospital for four sub-types of thyroid, e.g., thyroid adenoma (TA), follicular carcinoma (FC), follicular variant of PTC (FV-PTC), and medullary carcinoma (MC). It consists of 1,969 ultrasound images from 290 patients. Five-fold cross-validation is utilized for this dataset. In addition, a subset of the data, **FJTU-H** (349 images from FC and 174 images from FV-PTC), includes gender information and 9 heterogeneous attributes annotated by professional doctors. The attribute distribution exhibits severe heterogeneity within the same category as in Fig. 5. The FJTU-H is utilized for dataset analysis of heterogeneity and validation of heterogeneity cause C.

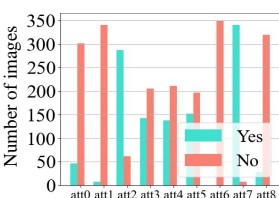

**Figure 5: The attribute distribution of FC category.**

## 5.2 Compared Methods and Evaluation Metrics

**Compared methods.** In order to comprehensively validate the effectiveness of MPCCI, we make comparisons with various methods. Initially, we select four representative deep learning models with backbone architectures of ResNet18 [18], VGG16 [40], ViT [9], and Mamba [15] for image processing. These four methods utilize distinct feature extraction mechanisms to identify image features, facilitating the assessment of MPCCI performance across different architectures. Additionally, we employ several representative supervised learning methods for image classification, such as CABNet [17] and CAD_PE [24], along with data augmentation techniques like MixupNet [51] and MixStyleNet [54], as well as invariant

**Table 1: Performance comparison between MPCCI and compared methods on the CT COVID-19 dataset. The best and second best results are marked in bold and with underline respectively.**

| Method | ACC (%) | P (%) | R (%) | F1 (%) |
|---|---|---|---|---|
| ResNet18[18] | 95.44±0.71 | **98.87**±0.94 | 92.36±1.03 | 95.50±0.36 |
| Fishr[39] | 96.06±1.30 | 94.57±0.28 | **97.31**±0.94 | 95.92±0.73 |
| CABNet[17] | 95.89±0.64 | 95.06±1.04 | 96.37±1.50 | 95.71±0.83 |
| MixupNet[51] | 95.10±0.45 | 96.82±1.51 | 92.74±1.89 | 94.74±0.57 |
| MixStyleNet[54] | 95.10±1.49 | 96.82±1.42 | 92.47±1.76 | 94.72±0.82 |
| VGG16[40] | 93.96±1.84 | 94.68±2.59 | 93.74±1.86 | 94.21±1.58 |
| ViT[9] | 95.22±0.21 | 95.59±0.58 | 94.52±0.71 | 95.06±0.63 |
| Mamba[15] | 79.67±0.00 | 77.38±0.00 | 86.50±0.00 | 81.69±0.00 |
| MPCCI (Ours) | **96.10**±0.51 | 96.18±0.24 | 96.37±0.84 | **96.28**±0.18 |

feature learning method Fishr [39], to further evaluate the performance of MPCCI. It is noteworthy that for the BUSI dataset, we also incorporate state-of-the-art modality-specific methods for breast cancer disease, including TNTs [16], BVA Net [46], HoVer-Trans [35], and MIB Net [42], for comparative analysis with MPCCI.

**Evaluation metrics.** We evaluate MPCCI using four commonly used metrics in classification tasks: accuracy (Acc), precision (P), recall (R), and F1-score (F1).

## 5.3 Experimental Details

In our experiments, we utilize ResNet18 as the backbone of MPCCI. All medical images are resized to 128 × 128 pixels. The model is trained using the SGD optimizer, with learning rates set to 0.0001/0.0001/0.001/0.0001 for the CT COVID-19, BUSI, FJPH, and FJTU datasets respectively. All experiments are conducted on a single Nvidia RTX3090 GPU, with batch sizes set to 10/10/128/10. The hyperparameter $\alpha$ is set to 0.5, while $\lambda$ is set to 0.1/0.1/1/0.1 for the four datasets respectively.

For the baseline methods, we reproduce the source codes of ResNet18 [18], VGG16 [40], ViT [9], Mamba [15], CABNet [17], MixupNet [51], MixStyleNet [54], Fishr [39], and CAD_PE [24]. All experimental results are based on the average of five experiments conducted with different random seeds. The results of TNTs [16], BVA Net [46], HoVer-Trans [35], and MIB Net [42] are directly cited from the original papers, as their source codes are not publicly accessible.

## 5.4 Experimental Results

Table 1, Table 2, and Table 3 present the overall performance of MPCCI and the compared methods. On CT COVID-19 dataset, our MPCCI method exhibits strong performance, underscoring its efficacy and robustness in COVID-19 image classification tasks. Specifically, it achieves the highest performances of 96.10% and 96.28% on precision and F1-score respectively. On the BUSI dataset, MPCCI achieves the best average results of four evaluation metrics, demonstrating its superiority. Compared to MIB Net, which takes advantages of multi-task learning method in both classification and segmentation, our MPCCI outperforms it with improvements of 0.26%, 0.99%, 2.94%, and 2.19% in accuracy, precision, recall, and F1-score, respectively. This demonstrates that our approach can

**Table 2: Performance comparison between MPCCI and compared methods on the BUSI and FJPH datasets for ultrasound images. The best and second best results are marked in bold and with underline respectively.**

| Method | BUSI | | | | FJPH | | | |
|---|---|---|---|---|---|---|---|---|
| | ACC (%) | P (%) | R (%) | F1 (%) | ACC (%) | P (%) | R (%) | F1 (%) |
| TNTs* [16] | 81.20±3.20 | 76.30±5.70 | 61.10±10.40 | 67.9±5.70 | - | - | - | - |
| BVA Net* [46] | 84.3 | 88.3 | 75.1 | - | - | - | - | - |
| HoVer-Trans* [35] | 85.50±5.00 | 87.60±6.20 | 86.70±11.50 | 87.20±8.00 | - | - | - | - |
| MIB Net* [42] | 92.97±1.11 | 93.21±1.50 | 92.97±1.10 | 92.85±1.01 | - | - | - | - |
| ResNet18 [18] | 91.39±2.48 | 91.76±1.72 | 95.91±1.82 | 93.75±1.78 | 80.90±1.12 | 80.31±1.17 | 87.60±4.40 | 83.74±0.87 |
| Fishr [39] | 93.04±2.74 | **96.61**±1.34 | 93.18±2.36 | 94.34±1.23 | 84.26±0.84 | **87.87**±2.91 | 74.35±3.77 | 80.55±2.45 |
| CABNet [17] | 89.23±5.37 | 95.12±1.78 | 88.63±3.63 | 91.76±3.15 | 83.14±1.49 | 83.33±4.63 | 76.92±9.92 | 80.00±7.17 |
| MixupNet [51] | 92.30±2.69 | 95.34±1.68 | 93.18±2.51 | 94.25±1.83 | 84.26±2.34 | 79.06±3.21 | 87.17±2.95 | 82.92±2.14 |
| MixStyleNet [54] | 86.15±5.04 | 90.69±4.93 | 88.63±6.42 | 89.65±4.64 | 76.40±1.18 | 76.47±2.43 | 66.66±3.88 | 71.23±2.94 |
| VGG16 [40] | 93.12±1.08 | 94.45±1.26 | 94.45±1.45 | 94.45±1.31 | 82.47±1.80 | 83.33±0.98 | 85.60±4.40 | 84.41±0.74 |
| ViT [9] | 90.76±3.72 | 93.18±3.55 | 93.18±4.16 | 93.18±3.77 | 74.83±0.45 | 74.33±0.67 | 84.40±1.60 | 79.03±0.21 |
| Mamba [15] | 67.69±0.00 | 67.69±0.00 | **100.00**±0.00 | 80.73±0.00 | 68.53±0.00 | 68.33±0.00 | 82.00±0.00 | 74.54±0.00 |
| MPCCI (Ours) | **93.23**±2.15 | 94.20±1.36 | 95.91±1.82 | **95.04**±1.59 | **85.62**±3.14 | 86.62±5.05 | **88.80**±3.20 | **87.57**±2.23 |

*Note: These results are directly cited from the original papers, as their source codes are not publicly accessible.

**Table 3: Performance comparison on FJTU. The best and second best results are marked in bold and with underline respectively.**

| Method | FV-PTC | FC | TA | MC |
|---|---|---|---|---|
| Fishr | 85.42 | 70.68 | 57.77 | 71.02 |
| CABNet | 86.99 | 71.30 | 58.03 | 71.73 |
| MixupNet | 86.79 | 65.15 | 56.63 | 70.03 |
| MixStyleNet | 86.86 | 62.28 | 57.45 | 66.33 |
| CAD_PE | 86.84 | 71.48 | 57.91 | 72.46 |
| ResNet18 | 86.05 | 68.98 | 57.52 | 74.67 |
| MPCCI | **87.15** | **72.77** | **59.26** | **77.99** |

perform well in diagnosing breast cancer in ultrasound images with only instance-level labels. On the FJPH dataset, the performance of MPCCI consistently surpasses the runner-up by 1.36%, 1.20%, and 3.83% in accuracy, recall, and F1-score, respectively. The results highlight its consistent superiority across various datasets and underscores its potential for practical applications in medical image analysis. On the FJTU dataset, we test the accuracy for the four sub-types. Compared to SOTA and baseline methods, the proposed MPCCI achieves the best performance across all categories. While our method generally outperforms the compared methods, it occasionally lags behind by 1% to 2% in Precision or Recall. This discrepancy may stem from our method's suboptimal performance in handling specific categories. A potential improvement direction is to leverage prototype learning for enhancing the classification boundaries [41].

**Ablation Study.** We conduct ablation studies on CT COVID-19 and FJPH datasets to explore the individual contributions of each component of the proposed MPCCI. The four components evaluated are MVFE, MPC, and its two contained criteria (i.e., IP & MT). The results of ablation studies are presented in Table 4 with AB1: ResNet18 (baseline), AB2: AB1+MVFE, AB3: AB2+MPC, AB4: AB3+IP, and AB5: MPCCI (AB4+MT). The results show that both

MVFE and MPC contribute to the improvements in performance compared to the baseline, demonstrating that MPCCI based on FDA can effectively evaluate the causal effect from image to label. Moreover, the results also reveal that it is crucial to consider both IP and MT factors simultaneously, as the performance drops significantly when only IP is considered.

**Validation of Heterogeneity Cause C.** The gender/age is the unobserved confounding factor C [1]. Inspired by this work [1], the FJTU-H dataset is divided into male and female groups. Subsequently, we conduct generalization tests on these two groups separately. The higher generalizability indicates that the algorithm is less influenced by the confounding factor of gender. The results are shown in Table 5. It can be seen that MPCCI is less affected by the confounding factor C (gender).

**Data Set Analysis.** We utilize the FJTU-H dataset to validate the effectiveness of our method in addressing heterogeneity. Two sets of control experiments are conducted: random splitting and splitting by low/high heterogeneity. The Pearson correlation in low and high heterogeneity groups are 0.8 and 0.5, respectively. The experimental results are shown in Table 6. It can be found that in the random splitting group, the baseline method (ResNet18) and our MPCCI method achieve similar results. However, in the splitting by heterogeneity group, our method significantly outperforms the baseline, demonstrating its effectiveness in handling the heterogeneity issue.

**Experimental Extensions.** In this section, we aim to address three questions to provide more detailed analysis of the proposed method: 1) What is the optimal number of expert networks required to optimize feature views? 2) How do the mixing mechanism and fusion module contribute to the performance of MPC? 3) Can the interpretability of MPCCI be quantitatively assessed? To answer the first question, we conduct experiments by changing the number of expert networks from one to nine on the FJPH dataset and observe the performance. The results in Fig. 6 (a) show that the optimal performance is achieved with five expert networks, indicating that an increased number of views does not correlate directly with

**Table 4: Ablation study of MPCCI on CT COVID-19 and FJPH. AB1: ResNet18 (baseline), AB2: AB1+MVFE, AB3: AB2+MPC, AB4: AB3+IP, and AB5: MPCCI (AB4+MT). The best and second best results are marked in bold and with underline respectively.**

| Method | CT COVID-19 | | | | FJPH | | | |
|---|---|---|---|---|---|---|---|---|
| | ACC(%) | P (%) | R (%) | F1 (%) | ACC(%) | P (%) | R (%) | F1 (%) |
| AB1 | 95.44±0.71 | **98.87**±0.94 | 92.36±1.03 | 95.50±0.36 | 80.90±1.12 | 80.31±1.17 | 87.60±4.40 | 83.74±0.87 |
| AB2 | 95.47±0.33 | 95.47±0.57 | 95.91±0.75 | 95.69±0.25 | 81.57±0.45 | 80.54±3.46 | 88.80±3.20 | 84.41±0.50 |
| AB3 | 95.72±1.13 | 98.20±1.61 | 93.54±1.34 | 95.81±1.42 | 84.95±2.69 | 83.93±5.87 | **90.40**±1.60 | 86.99±1.90 |
| AB4 | 95.92±0.47 | 97.68±1.21 | 94.47±1.13 | 96.05±0.24 | 83.37±0.90 | 82.36±0.97 | 89.60±0.40 | 85.82±0.71 |
| AB5 | **96.10**±0.51 | 96.18±0.24 | **96.37**±0.84 | **96.28**±0.18 | **85.62**±3.14 | **86.62**±5.05 | 88.80±3.20 | **87.57**±2.23 |

**Table 5: Results of generalization ability for the unobserved confounder C (gender) on FJTU-H dataset**

| Method | Male2Female | | Female2Male | |
|---|---|---|---|---|
| | ACC(%) | F1 (%) | ACC(%) | F1 (%) |
| ResNet18 | 70.43 | 78.96 | 66.55 | 77.02 |
| MPCCI | **74.20** | **85.16** | **67.21** | **77.88** |

**Table 6: Results of heterogeneous generalization on FJTU-H dataset**

| Method | Random Group | | Heterogeneous Group | |
|---|---|---|---|---|
| | ACC(%) | F1 (%) | ACC(%) | F1 (%) |
| ResNet18 | 93.67 | 95.43 | 73.41 | 81.29 |
| MPCCI | **93.74** | **95.68** | **77.63** | **83.29** |

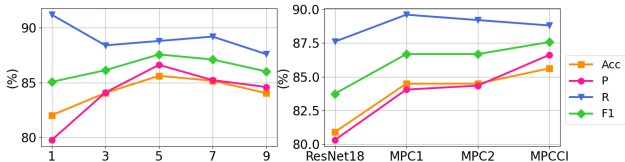

**Figure 6: (a) Performance change by increasing the number of expert networks in MVFE. (b) Effect exploration to the contributions of different mixing mechanisms and fusion modules in MPC.**

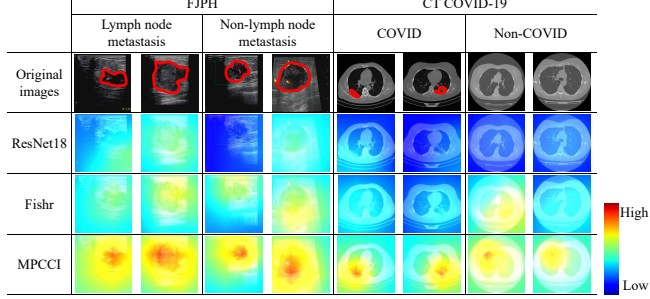

**Figure 7: Comparative visualization of lesion detection by ResNet18, Fishr, and MPCCI on FJPH and CT COVID-19 datasets. Professional doctors have annotated the lesions in the original images, which are delineated in red for clarity.**

enhanced performance. To address the second question, we assess the performance of "MPC1" mode (where the mixing mechanism is removed, and $\mathbf{A}$ is fused with only the source multi-view prototypes $\mathbf{S}^c$) and "MPC2" mode (where the fusion module is removed, and $\hat{\mathbf{A}}$ is calculated by randomly mixing $\mathbf{A}$ and $\mathbf{S}^c$). From Fig. 6 (b), we observe that both the mixing mechanism and the fusion module are necessary for MPC, demonstrating its ability to simulate the feature representation of lesions in various states. To answer the third question, we visualize the class activation maps (CAM) [53] for four samples in the FJPH and CT COVID-19 datasets using baseline (ResNet18), Fishr, and MPCCI in Fig. 7. The lesions in the original images have been marked by professional doctors and shown with red circles. By comparing the locations of the lesions and visualization results, we can see that MPCCI can attend to the entire lesion, while ResNet18 and Fishr can only focus on a small part of the lesion or miss it entirely. Therefore, the CAM results of MPCCI can provide interpretable results to doctors, thereby facilitating medical diagnosis.

## 6 Conclusion

In this paper, we propose a novel approach MPCCI for enhanced medical image classification by addressing the unmeasurable confounding factors present in medical imaging analysis. Leveraging FDA, MPCCI estimates the total causal effect of an image on its corresponding label, thus mitigating the negative effects of the confounders. The proposed approach comprises an MVFE module with spatial-channel attention, allowing multi-view features to serve as mediators in FDA, and an MPC module to effectively

apply causal intervention on the mediators. An adaptive training strategy, including IP and MT, is introduced to maintain the stable training during the feature exchange process. Experimental results on four medical datasets demonstrate the effectiveness of MPCCI, achieving high accuracy, precision, recall, and F1-score in diagnosing COVID-19, breast cancer, lymph node metastasis, and thyroid. In the future, we plan to conduct extensive validation studies across a wider spectrum of medical conditions and imaging modalities. By rigorously evaluating the performance of MPCCI on diverse datasets encompassing a myriad of medical scenarios, we aim to demonstrate its efficacy and versatility in facilitating accurate and reliable diagnostic decision-making.

## Acknowledgments

This work was supported by the Research Grants Council of the Hong Kong SAR (Grant No. PolyU11211521, PolyU15218622, PolyU15215623, PolyU25216423, and C5052-23G), The Hong Kong Polytechnic University (Project IDs: P0035379, P0043563, P0046094, and P0051130), and the National Natural Science Foundation of China (Grant No. 62202399, U21A20512, and 62306259).

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
