# OpenReview forum: "Mixed Prototype Correction for Causal Inference in Medical Image Classification"
_acmmm.org/ACMMM/2024/Conference — MM2024 Poster_

### Official Review · Reviewer_sqgJ · 2024-05-19

**Rating:** 4
**Confidence:** 3

**Summary:**

This paper proposes a causal inference-based mixed prototype correction model to improve classification performance by reducing the influence of confounding factors on the causal relationship between image features and labels. A multi-view feature extraction module is used to establish mediators, and a mixed prototype correction model is employed for causal intervention. To ensure the stability of the model, an adaptive training startegy is proposed. It is the first attempt to address the impact of medical image heterogeneity on disease diagnosis. The experimental results demonstrate the contributions of each module to the model's performance, and compared to other methods, the proposed approach achieves significant improvements.

**Strengths:**

The method takes a causal analysis perspective, which is innovative in the field of medical image analysis.
The method achieves good performance on multiple datasets with different types of diseases, indicating a certain level of generalization ability.
The MVFE module with spatial-channel attention mechanism enhances the model's representational power. And the MPC module  improves the classification accuracy.
The Class Activation Mapping (CAM) visualization provides the interpretability of the model.

**Limitations:**

The limitations of the method should be further discussed.
The performance improvement on the CT COVID-19 and BUSI datasets is small.
It may need to further validate on a broader range of disease types and imaging modalities.
The paper  dose not provide the source code for  other researchers to replicate and verify the study's results.

**Suitability:**

3

---

### Official Review · Reviewer_nJeo · 2024-05-22

**Rating:** 6
**Confidence:** 3

**Summary:**

This work introduces a novel medical image classification model, the Mixed Prototype Correction for Causal Inference (MPCCI). To enhance interpretability, the model employs the front-door adjustment scheme to clarify the causal relationship between medical images and disease labels. An adaptive training strategy is implemented using the Information Purity (IP) and Maturity metrics to stabilize the training process. Numerical experiments on three medical image datasets demonstrate that MPCCI achieves high diagnostic accuracy.

**Strengths:**

- MPCCI offers a novel approach to interpreting medical image classification models.
- The adaptive training strategy ensures stable model training.
- Ablation studies demonstrate performance improvements with the front-door adjustment (FDA) framework.

**Limitations:**

- Some of the methods compared in the experiments are outdated.
- While the MPCCI model shows performance improvement on the FJPH dataset classification, it does not consistently outperform all compared methods across other datasets.

**Suitability:**

2

---

### Official Review · Reviewer_iVA6 · 2024-05-24

**Rating:** 3
**Confidence:** 3

**Summary:**

The present paper introduces a novel method named Mixed Prototype Correction for Causal Inference (MPCCI) to address challenges related to image heterogeneity in medical image classification. By integrating causal inference with adaptive training strategies, this method aims to mitigate the impact of unseen confounding factors and enhance the accuracy of deep learning models in disease diagnosis.
In the experimental section, the authors evaluated the performance of MPCCI on three medical image datasets, including CT and ultrasound image modalities. The experimental results demonstrate that MPCCI exhibits superior diagnostic accuracy and reliability in diagnosing diseases such as COVID-19, breast cancer, and lymph node metastasis. The method outperformed comparative approaches in terms of accuracy, precision, recall, and F1 score.

**Strengths:**

#### Accurate Problem Identification
The paper effectively addresses the issues of heterogeneity and unseen confounding factors in medical image classification. By presenting a targeted solution, it holds significant potential for improving the accuracy of disease diagnosis.

#### Methodological Innovation
By incorporating causal inference theory, specifically the front-door adjustment strategy, to address confounding factors in medical image analysis, the study introduces a novel approach that offers a fresh perspective on the field.

#### Rigorous Experimental Design
The authors conducted extensive experimental validation across multiple datasets, including different modalities of medical images. This rigorous approach enhances the credibility of the research findings.

#### Superior Performance
Compared to existing techniques, MPCCI demonstrates superior performance across multiple evaluation metrics, including accuracy, precision, recall, and F1 score.

#### Clinical Application Potential
The proposed method shows promising performance in diagnosing various diseases, indicating its potential application in clinical practice.

#### Excellent Results Presentation
In terms of results visualization, the paper provides clear and informative images that vividly illustrate the advantages of the proposed method and its focus on lesion areas. This makes the data presentation exceptionally transparent and easy to understand.

**Limitations:**

#### Incomplete Comparative Experiments
The study lacks comprehensive comparative experiments regarding the selection of CNN structures in the MCFE module. The paper primarily presents results based on ResNet18 without comparing them to other CNN architectures. It is crucial to explain whether the choice of feature maps impacts subsequent results and to justify why ResNet18 was ultimately selected as the base network for feature extraction.

#### Insufficient Data Explanation
Given that the paper's primary aim and innovation lie in integrating causal inference and adaptive training strategies to mitigate the impact of unseen confounding factors and enhance the accuracy of deep learning models in disease diagnosis, it fails to demonstrate whether the three datasets used actually encompass these issues. For instance, in the CT COVID-19 dataset, are there significant variations in CT images among infected individuals? The paper does not provide detailed information on this aspect, making it difficult for readers to assess whether the proposed method effectively addresses these problems.

#### Performance Limitations
While the method proposed in the paper performs well across various metrics, it is not always the best compared to other models. There are instances where other models outperform the proposed method in specific metrics.

#### Hyperparameter Selection
The paper does not offer detailed information on hyperparameter selection, such as learning rate, batch size, and choice of optimizer. It is essential to explain how these hyperparameters were tuned and determined.

#### Explanation of Heterogeneity Cause C
The paper mentions the heterogeneity cause C, but it remains unclear whether this C is automatically learned by the model or inherent in the data. Moreover, there is no professional validation of this cause C. It is necessary to determine whether C is accurately identified, its accuracy, and its effectiveness. Incorporating visual outputs would enhance the interpretability of the model's structure.

### Specific Questions and Recommendations

1. **Comparative Analysis of CNN Structures**: Have comparative experiments been conducted to evaluate different CNN structures within the MCFE module? The paper only shows results for ResNet18 but lacks comparisons with other CNN architectures. Please provide an explanation of whether the feature maps influence the subsequent results and justify why ResNet18 was chosen as the base network for feature extraction.

2. **Data Set Analysis**: Does the paper sufficiently address whether the three datasets used exhibit issues related to heterogeneity and unseen confounding factors? For example, in the CT COVID-19 dataset, are there significant differences in CT images among infected individuals? The paper should detail this aspect to help readers evaluate the method's effectiveness in addressing these problems.

3. **Performance Metrics**: While the proposed method performs well across various metrics, it is not consistently the best. Can you provide a more detailed comparison and analysis of where other models outperform your method and discuss potential improvements?

4. **Hyperparameter Details**: The paper should include detailed information on the selection and tuning of hyperparameters, such as learning rate, batch size, and optimizer choice. How were these hyperparameters adjusted to optimize performance?

5. **Validation of Heterogeneity Cause C**: The paper mentions a heterogeneity cause C but lacks clarity on whether it is model-generated or data-inherent. There is no professional validation of this cause. Could you elaborate on the accuracy and effectiveness of identifying C, and possibly include visual outputs to enhance the model's interpretability?

**Suitability:**

2

---

### Meta-Review · Area_Chair_vVfZ · 2024-06-28

**Recommendation:** Accept (Poster)
**Confidence:** 5

**Metareview:**

The paper introduces the Mixed Prototype Correction for Causal Inference (MPCCI), a novel method addressing challenges related to image heterogeneity in medical image classification. By integrating causal inference with adaptive training strategies, the method aims to mitigate the impact of unseen confounding factors and enhance diagnostic accuracy. The experimental evaluation on three medical image datasets demonstrates the effectiveness of MPCCI in diagnosing diseases such as COVID-19, breast cancer, and lymph node metastasis.
Despite some limitations, such as the need for more comprehensive comparative experiments and detailed explanations of hyperparameter selection, the method demonstrates significant potential for clinical application. The strengths of the paper, including its methodological innovation, rigorous experimental design, and superior performance, outweigh the limitations. The authors have effectively addressed most of the concerns raised during the review process, making this submission a valuable contribution to the field and suitable for presentation as a poster at ACM Multimedia 2024.